# A Simultaneous Localization and Mapping (SLAM) Framework for 2.5D Map Building Based on Low-Cost LiDAR and Vision Fusion

**Guolai Jiang** [1,2,3]**, Lei Yin** [3]**, Shaokun Jin** [1,2,3]**, Chaoran Tian** [1,2]**, Xinbo Ma** [2]
**and Yongsheng Ou** [1,3,4,*] 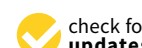

1   Shenzhen Institutes of Advanced Technology, Chinese Academy of Sciences, Shenzhen 518055, China
2   Shenzhen College of Advanced Technology, University of Chinese Academy of Sciences,
    Shenzhen 518055, China; cr.tian@siat.ac.cn (C.T.); xb.ma@siat.ac.cn (X.M.)
3   Guangdong Provincial Key Laboratory of Robotics and Intelligent System, Shenzhen Institutes of
    Advanced Technology, Chinese Academy of Sciences, Shenzhen 518055, China; sk.jin@siat.ac.cn (S.J.);
    gl.jiang@siat.ac.cn (G.J.); lei.yin@siat.ac.cn (L.Y.)
4   The CAS Key Laboratory of Human-Machine-Intelligence Synergic Systems, Shenzhen Institutes of
    Advanced Technology, Chinese Academy of Sciences, Shenzhen 518055, China
*   Correspondence: ys.ou@siat.ac.cn

**Abstract:** The method of simultaneous localization and mapping (SLAM) using a light detection and ranging (LiDAR) sensor is commonly adopted for robot navigation. However, consumer robots are price sensitive and often have to use low-cost sensors. Due to the poor performance of a low-cost LiDAR, error accumulates rapidly while SLAM, and it may cause a huge error for building a larger map. To cope with this problem, this paper proposes a new graph optimization-based SLAM framework through the combination of low-cost LiDAR sensor and vision sensor. In the SLAM framework, a new cost-function considering both scan and image data is proposed, and the Bag of Words (BoW) model with visual features is applied for loop close detection. A 2.5D map presenting both obstacles and vision features is also proposed, as well as a fast relocation method with the map. Experiments were taken on a service robot equipped with a 360° low-cost LiDAR and a front-view RGB-D camera in the real indoor scene. The results show that the proposed method has better performance than using LiDAR or camera only, while the relocation speed with our 2.5D map is much faster than with traditional grid map.

**Keywords:** SLAM; localization; mapping; relocation

## 1. Introduction

Localization and navigation are the key technologies of autonomous mobile service robots, and simultaneous localization and mapping (SLAM) is considered as an essential basis for this. The main principle of SLAM is to detect the surrounding environment through sensors on the robot, and to construct the map of the environment while estimating the pose (including both location and orientation) of the robot. Since SLAM was first put forward in 1988, it was growing very fast, and many different schemes have been formed. Depending on the main sensors applied, there are two mainstream practical approaches: LiDAR (light detection and Ranging)-SLAM and Visual-SLAM.

### 1.1. LiDAR-SLAM

LiDAR can detect the distance of the obstacles, and it is the best sensor to construct a grid map, which represents the structure and obstacles on the robot running plane. The early SLAM research

often used LiDAR as the main sensor. Extended Kalman filter (EKF) was applied to estimate the pose and of the robot [1], but the performance was not ideal. For some strong nonlinear systems, this method will bring more truncation errors, which may lead to inaccurate positioning and mapping. Particle filter approaches [2,3] were introduced because they can effectively avoid the nonlinear problem, but it also leads to the problem of increasing the amount of calculation with the increase of particle number. In 2007, Grisetti proposed a milestone of LiDAR-SLAM method called Gmapping [2], based on improved Rao-Blackwellized particle filter (RBPF), it improves the positioning accuracy and reduces the computational complexity by improving the proposed distribution and adaptive re-sampling technique.

As an effective alternative to probabilistic approaches, optimization-based methods are popular in recent years. In 2010, Kurt Konolige proposed such a representative method called Karto-SLAM [3], which uses sparse pose adjustment to solve the problem of matrix direct solution in nonlinear optimization. Hector SLAM [4] proposed in 2011 uses the Gauss-Newton method to solve the problem of scanning matching, this method does not need odometer information, but high precision LiDAR is required. In 2016, Google put forward a notable method called Cartographer [5], by applying the laser loop closing to both sub-maps and global map, the accumulative error is reduced.

### 1.2. Visual-SLAM

Using visual sensors to build environment map is another hot spot for robot navigation. Compared with LiDAR-SLAM, Visual-SLAM is more complex, because image carries too much information, but has difficulty in distance measurement. Estimating robot motion by matching extracted image features under different poses to build a feature map is a common method for Visual-SLAM.

Mono-SLAM [6], proposed in 2007, is considered the origin of many Visual-SLAM. Extended Kalman filter (EKF) is used as the back-end to track the sparse feature points in the front-end. The uncertainty is expressed by a probability density function. From the observation model and recursive calculation, the mean and variance of the posterior probability distribution are obtained. Reference [7] used RBPF to realize visual-SLAM. This method avoids the problem of non-linear and has high precision, but it needs a large number of particles, which increase the computational complexity. PTAM [8] is a representative work of visual-SLAM, it proposed a simple and effective method to extract key frames, as well as a parallel framework of a real-time tracking thread and a back-end nonlinear optimization mapping thread. It is the first time to put forward the concept of separating the front and back ends, leading the structure design of many SLAM methods.

ORB-SLAM [9] is considered as a milestone of visual-SLAM. By applying Oriented FAST and Rotated BRIEF (ORB) features and bag-of-words (BOW) model, this method can create a feature map of the environment in real-time stably in many cases. Loop detection and closing via BOW is an outstanding contribution of this work, it effectively prevents the cumulative error and can be quickly retrieved after the tracking lost.

In recent years, different from feature-based methods, some direct methods of visual-SLAM were explored, by estimating robot motion through pixel value directly. Dense image alignment based on each pixel of the images proposed in Reference [10] can build a dense 3D map of the environment. The work in Reference [11] built a semi-dense map by estimating the depth values of pixels with a large gradient in the image. Engel et al. proposed LSD-SLAM (Large-Scale Direct Monocular SLAM) algorithm [12]. The core of LSD-SLAM algorithm is to apply a direct method to semi-dense monocular SLAM, which is rarely seen in the previous direct method. Forster et al. proposed SVO (Semi-Direct Monocular Visual Odometry) [13] in 2014, called "sparse direct method", which combines feature points with direct methods to track some key points (such as corners, etc.), and then estimates the camera motion and position according to the information around the key points. This method runs fast for Unmanned Aerial Vehicles (UAV), by adding special constraints and optimization to such applications.

RGB-D camera can provide both color and depth information in its view field. It is the most capable sensor for building a complete 3D scene map. Reference [14] proposes Kinect fusion method, which uses the depth images acquired by Kinect to measure the minimum distance of each pixel in each frame, and fuses all the depth images to obtain global map information. Reference [15] constructs error function by using photo-metric and geometric information of image pixels. Camera pose is obtained by minimizing the error function. Map problem is treated as pose graph representation. Reference [16] is a better direct RGB-D SLAM method. This method combines the intensity error and depth error of pixels as error functions, and minimizes the cost function to obtain the optimal camera pose. This process is implemented by g2o. Entropy-based key frame extraction and closed-loop detection method are both proposed, thus greatly reducing the path error.

### 1.3. Multi-Sensor Fusion

Introducing assistant sensor data can improve the robustness of the SLAM system. Currently, for LiDAR-SLAM and Visual-SLAM, the most commonly used assistant sensors are encoder and Inertial Measurement Unit (IMU), which can provide additional motion data of the robot. SLAM systems [2,5,17–20] with such assistant sensors usually perform better.

In recent years, based on the works of LiDAR-SLAM and Visual-SLAM, some researchers have started to carry out the research of integrating such two main sensors [21–25]. In [21], the authors applied a visual odometer to provide initial values for two-dimensional laser Iterative Closets Point (ICP) on a small UAV, and achieved good results in real-time and accuracy. In [22], a graph structure-based SLAM with monocular camera and laser is introduced, with the assumption that the wall is normal to the ground and vertically flat. [23] integrates different state-of-the art SLAM methods based on vision, laser and inertial measurements using an EKF for UAV in indoor. [24] presents a localization method based in cooperation between aerial and ground robots in an indoor environment, a 2.5D elevation map is built by RGB-D sensor and 2D LiDAR attached on UAV. [25] provides a scale estimation and drift correction method by combining mono laser range finder and camera for mono-SLAM. In [26], a visual SLAM system that combines images acquired from a camera and sparse depth information obtained from 3D LiDAR is proposed, by using the direct method. In [27], EKF fusion is performed on the poses calculated by LiDAR module and vision module, and an improved tracking strategy is introduced to cope with the tracking failure problem of Vision SLAM. As camera and LiDAR becoming standard configurations for robots, laser-vision fusion will be a hot research topic for SLAM, because it can provide a more robust result for real applications.

### 1.4. Problems in Application

Generally speaking, LiDAR-SLAM methods build occupy grid map, which is ready for path-planning and navigation control. However, closed-loop detection and correction are needed for larger map building, and that is not easy for a grid map. Because the scan data acquired are two-dimensional point cloud data, which have no obvious features and are very similar to each other, the closed-loop detection based on scan data directly is often ineffective. And this flaw also extends to fast relocation function when robot running with a given map. In the navigation package provided by the Robot Operating System (ROS), the robot needs a manually given initial pose before automotive navigation and moving. On the other hand, Most Visual-SLAM approaches generate feature maps, which are good at localization, but not good for path-planning. RGB-D or 3D-LiDAR are capable of building a complete 3D scene map, but it is limited in use because of high calculating or founding cost.

For consumer robots, the cost of sensors and processing hardware is sensitive. Low-cost LiDAR sensors become popular. However, to realize a robust low-cost navigation system is not easy work. Because low-cost LiDAR sensors have much poorer performance in frequency, resolution and precision than normal ones. In one of our previous work [28], we have introduced methods to improve the performance of scan-matching for such low-cost LiDAR, however, that only works for adjacent poses. The accumulating errors may usually grow too fast and cause failure to larger map building. To find

an effective and robust SLAM and relocation solution with low computation cost and the low founding cost is still a challenging work for commercially-used service robots.

*1.5. The Contributions of this Paper*

This paper proposes a robust low-cost SLAM frame work, with the combination of low-cost LiDAR and camera. By combining the laser points cloud data and image feature points data as constraints, a graph optimization method is used to optimize the pose of the robot. At the same time, the BoW based on visual features is used for loop closure detection, and then the grid map built by laser is further optimized. Finally, a 2.5D map presenting both obstacles occupy and vision feature is built, which is ready for fast re-localization.

The main contribution/advantages of this paper are:

- Proposes a new low-cost laser-vision slam system. For graph optimization, a new cost-function considering both laser and feature constraints. We also added image feature-based loop-closure to the system to remove accumulative errors.
- Proposes a 2.5D map, it is fast in re-localization and loop-closing, compared with feature map, it is ready for path-planning.

The remaining parts of this paper are organized as follows: Section 2 introduces the slam framework based on graph optimization; Section 3 introduces the back-end optimization and loop closing method; Section 4 introduces the 2.5D map and fast relocation algorithm; Section 5 is the experiment; and Section 6 is the conclusion part.

## 2. SLAM Framework Based on Graph Optimization

A graph-based SLAM framework could be divided into two parts: Front-end and back-end, as shown in Figure 1 below.

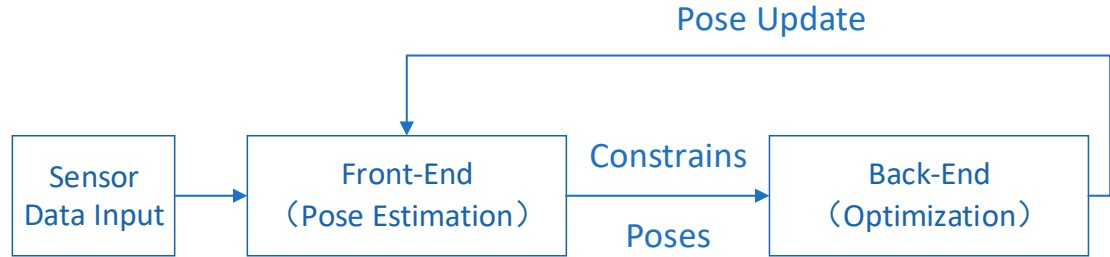

**Figure 1.** A general graph-based light detection and ranging (SLAM) framework.

The front-end is mainly used to estimate the position and pose of the robot by sensor data. However, the observed data contain varying degrees of noise whether in images or in laser. Especially for low-cost LiDAR and cameras. The noise will lead to cumulative errors in pose estimation, and such errors will increase with time. Through filtering or optimization algorithms, the back-end part can eliminate the cumulative errors and improve the positioning and map accuracy.

In this paper, graph optimization is used as the back-end, and the error is minimized by searching the descending gradient through nonlinear optimization. Graph optimization simply describes an optimization problem in the form of a graph. The node of the graph represents the position and attitude, and the edge represents the constraint relationship between the position and the attitude and the observation. The sketch map of graph optimization is shown in Figure 2.

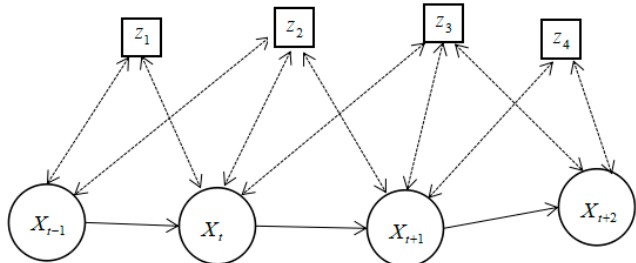

**Figure 2.** Sketch map of graph optimization.

In Figure 2, **X** represents the position and pose of the robot, and **Z** represents the observation. In this paper, the robot observes the external environment through both LiDAR and camera. As a result, **Z** could be a combination of three-dimensional spatial feature points detected by camera or obstacle points detected by LiDAR. Matching result and errors of visual feature points and scan data are an essential part to generate the nodes and constrain for graph optimization.

For the matching of visual feature points, the error is usually represented by re-projection error. The calculation of re-projection error needs to give two cameras corresponding to adjacent frames, two-dimensional coordinates of matched feature points in two images and three-dimensional coordinates of corresponding three-dimensional space points.

As shown in Figure 3, for the feature point pairs $p_1$ and $p_2$ in adjacent frames, a real-world spatial point $P$ corresponding to $p_1$ could be localized at first, and then $P$ is re-projected onto the latter frame to form the feature point position $\hat{p}_2$ of the image. Due to the error of position and pose estimation, the existence of $\hat{p}_2$ and $p_2$ is not a coincidence. The distance between these two points could be denoted as a re-projection error. Pure vision SLAM usually extracts and matches feature points, approaches like EPnP [29] can be applied to estimate the pose transformation $(R, t)$ of adjacent frames. Where $R$ represents the rotation matrix, and $t$ represents the transformation matrix.

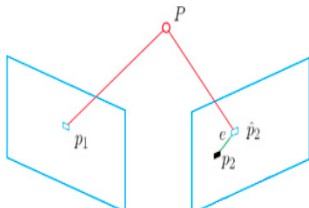

**Figure 3.** The re-projection error.

According to Figure 3, the re-projection errors could be calculated as follows:

Firstly, calculate the 3D position $[X', Y', Z']$ of $P'$ (corresponding point of $P$) in camera coordinate, through the pose and transformation relation $[\mathbf{R}, \mathbf{t}]$:

$$P' = RP + t = [X', Y', Z']^T. \tag{1}$$

Then, get the normalized coordinate $\mathbf{P}_c$:

$$\mathbf{P}_c = [u_c, v_c, 1]^T = [X'/Z', Y'/Z', 1]^T. \tag{2}$$

Based on camera parameters $(f_x, f_y, c_x, c_y)$, the re-projected pixel coordinate $\hat{p}_2 = (u_s, v_s)$ could be calculated as:

$$\begin{cases} u_s = f_x u_c + c_x \\ v_s = f_y v_c + c_y \end{cases}. \tag{3}$$

Finally, if there are $n$ feature points matched between two frames, the re-projection error is:

$$e_1 = \frac{1}{2} \sum_{i=1}^{n} \left\| p_i - \hat{p}_i' \right\|_2^2. \tag{4}$$

Compared with the visual error, the laser scan matching error is much simpler to obtain. LiDAR-SLAM usually needs scan matching to realize the estimation of pose transformation of adjacent frames. The estimated transformation $(R, t)$ cannot guarantee that all the laser data in the previous frame completely coincide with the laser data in the latter frame according to the pose transformation. The error could be calculated as:

$$e_2 = \sum_{i=1}^{n} \left\| M_i - (RS_i + t) \right\|_2^2, \tag{5}$$

where $n$ is the number of matched scan points, $M$ is the source scan point set and $S$ is the scan point set of an adjacent frame.

With multiple adjacent frame pairs get by Visual-SLAM or LiDAR-SLAM, the united re-projections errors of different frame pairs could be minimized for optimizing multiple poses.

## 3. The SLAM Framework of Low-Cost LiDAR and Vision Fusion

Base on the framework and key parts introduced in Section 2, this paper proposes a new SLAM framework of low-cost LiDAR and vision fusion. In the framework, for the back-end graph optimization, a new united error function of combining visual data matching error and laser data matching error is introduced. Meanwhile, in order to solve the problem of loop detection of traditional LiDAR-SLAM, a loop detection method is added, by introducing and visual data and bag-of-words. The integrated SLAM framework combining with laser and vision data is shown in Figure 4 below:

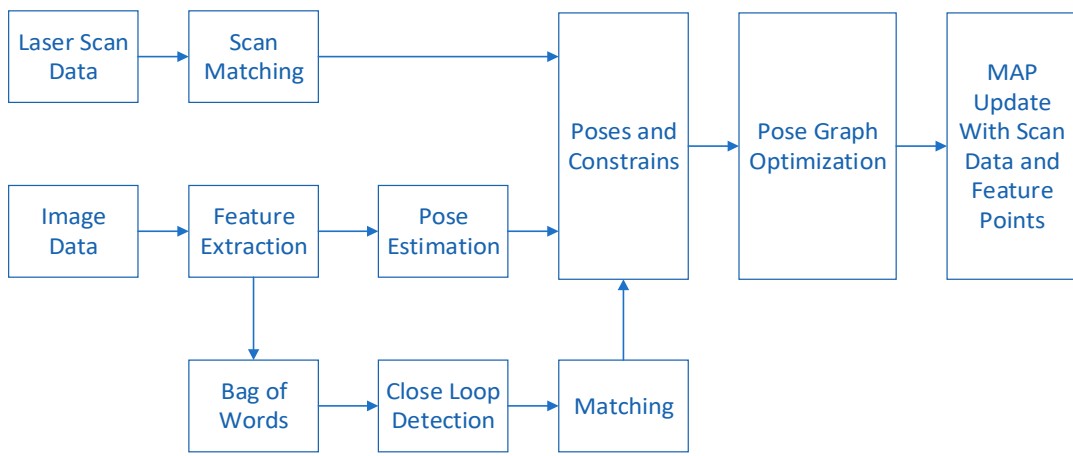

**Figure 4.** The proposed SLAM framework.

In the framework, both the laser data and image data can get the robot pose estimation. For laser data, scan-to-scan or scan-to-map methods can be applied to estimate current robot pose. For image data, we can use the ORB feature for image feature detection and generate the bag of words. Methods like EPnP [24] could be applied to estimate the pose transformation of adjacent frames.

### 3.1. United Error Function

The traditional error function for the matching of image data or scan data is given in Section 2. In this part, for further optimization and mapping procedure, we introduce a united error function with the fusion of image data and scan data.

Considering two adjacent robot poses with an initial guess of translation and rotation parameters [**R,t**], which could get through scan matching, image feature matching or even encoder data individually. Based on the error formulas introduced in Section 2, the united error function could be expressed as:

$$e = \beta \sum_{i=1}^{m} \left\| p_i - \frac{1}{Z_{i-1}} \mathbf{C}(RP_{i-1} + t)) \right\| + (1-\beta)\alpha \sum_{j=1}^{n} \left\| (P_j - (RP_{j-1} + t)) \right\|, \tag{6}$$

where $m$ is the number of matched features; $n$ is the number of corresponding scan points; $\mathbf{C}$ is the camera parameter matrix; $p_i$ is the image position of feature; $P_j$ is the scan point position; $\alpha$ is a parameter for unit conversion of the image pixel error and distance error, it is mainly judged by camera resolution and laser ranging range; $\beta$ is the parameter for balancing the two measurements, it takes value between (0,1). It is mainly judged by the two sensors' reliability and precision, for example, if the robot is working in an environment full of image features, $\beta$ can be larger so that we consider visual input more important. In contrast, if the scene is better for laser processing, $\beta$ can be smaller.

$(R, t)$ can be written in an algebra form, and the Lie Algebraic transformation formula is:

$$RP + t = \exp(\hat{\xi})P. \tag{7}$$

Then the error function could be reformulated as a Lie algebra form:

$$e = F(\xi) = \beta \sum_{i=1}^{m} \left\| p_i - \frac{1}{Z_{i-1}} \mathbf{C} \exp(\hat{\xi})P_{i-1}) \right\| + (1-\beta)\alpha \sum_{j=1}^{n} \left\| (P_j - \exp(\hat{\xi})P_{j-1}) \right\| \tag{8}$$

It is a function with the variable, $\xi$. For $K$ pairs of multiple poses with their transformation relations, the total error could be written as:

$$E = \sum_{i=1}^{K} e_i = \sum_{i=1}^{K} F_i(\xi_i), \tag{9}$$

and such error can be minimized to find better transformations and rotations between multiple poses.

### 3.2. Pose Graph Optimization

Due to the errors and noise of sensor measurements, the best matching of two adjacent poses may not be the best for the whole map. By unitizing a series of relative poses, the pose graph optimization could be applied for minimizing the errors during the SLAM process. In our approach, the robot pose set $x$ is regarded as the variable to be optimized, and the observations are regarded as the constraints between the poses.

The robot pose set $x$ is:

$$x = [x_1, x_2, x_3, \ldots, x_k]^T. \tag{10}$$

Based on the Lie Algebraic transformation set $\xi$, robot pose set $x$ is related to transformation set of $\xi$ by:

$$x = h(\xi). \tag{11}$$

Which also could be written as:

$$\xi = h^{-1}(x). \tag{12}$$

As a result, the total error of $K$ pairs of related poses could be rewritten as:

$$E = \sum_{i=1}^{K} e_i = \sum_{i=1}^{K} F_i(\xi_i) = f(x). \tag{13}$$

The target of pose graph optimization is to find a proper set $x$, so that the error $f(x)$ is minimized. Correspondingly, $\Delta x$ is the increment of the $x$ of the global independent variable. Therefore, when the increment is added, the objective function is:

$$\frac{1}{2}\left\| f(x + \Delta x) \right\|^2 \approx \frac{1}{2}\sum_{t=1}^{k}\left\| f(x_t) + J_t\Delta x_t \right\|^2,\tag{14}$$

where $J$ represents the Jacobian matrix, which is the partial derivative of the cost function to the variables, $k$ is the number of poses to be optimized, and the number of postures between the current frame and the loop frame is the global optimization.

Position and pose optimization can be regarded as the least square problem. The commonly used methods to solve the least square problem are Gradient Descent method, Gauss Newton method [30] and Levenberg-Marquadt (L-M) method [31]. The L-M method is a combination of Gadient Descent method and Gauss Newton method. This paper uses the L-M method to solve the least square problem.

According to L-M method and formula (14), in each iteration step, we need to find:

$$\min_{\Delta x_i}\frac{1}{2}\left\| f(x_i) + J(x_i)\Delta x_i \right\|^2 \text{ s.t. } \left\| I\Delta x_i \right\|^2 \le \mu,\tag{15}$$

where $\mu$ is the radius of the searching area (here we judge the area is a space sphere) in each step. The problem could be changed to solve the following incremental equation:

$$(J(x)^T J(x) + \lambda I)\Delta x = -J(x)^T f(x),\tag{16}$$

where $J$ represents the Jacobian matrix, and $\lambda$ is the Lagrange multiplier. As in each iteration step, $x$ is known, let $\mathbf{H} = J(x)^T J(x)$ and $\mathbf{g} = -J(x)^T f(x)$, it could be rewritten as:

$$(\mathbf{H} + \lambda\mathbf{I})\Delta x = \mathbf{g}.\tag{17}$$

As L-M method could be regarded as a combination of Gadient Descent method and Gauss Newton method in the formula (17), it could be more robust to get a reliable $\Delta x$ even if $\mathbf{H}$ is not invertible. By solving the above linear equation, we can obtain $\Delta x$, and then iterate $\Delta x$ to obtain $x$ corresponding to the minimum objective function, that is robot pose. Generally speaking, the dimension of $\mathbf{H}$ is very large, and the complexity of matrix inversion is $O(n^3)$. However, because $\mathbf{H}$ contains the constraints between poses, and only the adjacent posed have direct constraints, most of the elements are 0. $\mathbf{H}$ is sparse and shaped like an arrow. The calculation of (17) could be accelerated.

In practice, the pose graph optimization problem could be solved through tools like g2o [32] or ceres [33], with given error function and initial value. In our work, we apply g2o to solve this problem. The error function is formula (8), and the initial value of $x$ is obtained mainly through scan matching.

### 3.3. Loop Detection

Loop detection is a core problem in SLAM. Recognizing the past places and adding loop pose constraints to the pose graph can effectively reduce the cumulative error and improve the positioning accuracy. SLAM algorithms based on LiDAR are often unable to detect the loop effectively, because scan data can only describe the obstacle structure on the LiDAR installation plane, which usually lack of unique features of the scene. In other words, there may be multiple places that have very similar scan data to be detected, such as long coordinator, office workplace card, rooms with similar structure, etc. The rich texture features of visual images can just make up for this defect in our work.

In this paper, BoW is applied to construct the dictionary corresponding to the key-frames by visual features. Because the number of images captured by building indoor maps is too large, and the adjacent images have a high degree of repeatably, the first step is to extract the key-frames. The key

frame selection mechanism is as follows: (1) It has passed 15 frames since the last global relocation; (2) the distance from the previous key-frame has been 15 frames. (3) key frames must have tracked at least 50 three-dimensional feature points.

Among them, (1) (2) is the basis of its uniqueness, because in a short period of time the characteristics of the field of vision will not change significantly (3) to ensure its robustness, too few map points will lead to an uneven calculation error.

For each key-frame, the ORB image features and their visual words in BoW dictionary are calculated and saved as a bag of features. As a result, a key-frame stores the robot pose and bag of features, as well as the scan data obtained in that pose. It should be denoted that the key-frame here is applied for loop detection or relocation, the robot pose of each key-frame is updated after global optimization.

During the loop detection procedure, for an incoming image frame, the bag of feature is calculated and matched with all possible previous key-frames (possible means the key-frame should be a period of time before, and close enough in an estimated pose during SLAM procedure). When the current image frame is sufficiently similar to a key frame image by match the bag of feature, closing loop possibility is considered. Then, the current frame and that key frame are matched by both the scan data and image data, and the error in formula (6) is also calculated. If the matching error is small enough, the loop is confirmed and the matching result is added as constraints to the graph optimization framework, so that the accumulative errors can be eliminated.

## 4. 2.5D Map Representation and Relocation

In this part, we introduce a new expression of the 2.5D map, based on the scan data and visual data collected during the proposed method, as well as a fast relocation approach via the map.

### 4.1. Traditional Grid Map and Feature Map

Occupy grid map is wildly used in LiDAR-SLAM, it is a simple kind of 2D map that represents the obstacles on the LiDAR plane:

$$M_{grid} = \{m_g(x, y)\}, \tag{18}$$

where $m_g(x, y)$ denotes the possibility if the grid $(x, y)$ is occupied. Generally, the value of $m_g(x, y)$ can be 1 (the grid $(x, y)$ is occupied) or 0 (the grid $(x, y)$ is not occupied).

Feature map is another kind map generated by most feature-based Visual-SLAM approaches, it can be represented as:

$$M_{feature} = \{f(x, y, z)\}, \tag{19}$$

where $f(x, y, z)$ denotes that on the world position $(x, y, z)$, there is a feature $f(x, y, z)$, for real applications, $f(x, y, z)$ could be a descriptor to the feature in a dictionary like BoW.

While Visual-SLAM processing, key-frames with bag-of-feature description and poses at where they were observed are usually stored as an accessory for feature maps. For relocation applications, searching and matching in key-frames can greatly improve the processing speed and reliability.

As a result, a feature map is a sparse map which only has value on the position which has features. This makes a feature map is reliable for localization, but not good for navigation and path-planning.

### 4.2. Proposed 2.5D Map

Based on previous sections, the obstacle detected in the 2D laser plane and the image features detected by the camera can be aligned in one map, which can be represented as:

$$M = \{m(x, y)\}, \tag{20}$$

$$m(x, y) = \{m_g(x, y), feature\_list(x, y)\}, \tag{21}$$

where $feature\_list(x, y)$ represents the features in the corresponding vertical space of the grid $(x, y)$. If there is no feature detected above grid $(x, y)$, $feature\_list(x, y) = null$.

A general form of $feature\_list(x, y)$ could be:

$$feature\_list(x, y) = \{f\_list(x, y, z_1), f\_list(x, y, z_2), \ldots, f\_list(x, y, z_n)\}, \qquad (22)$$

where $z_1, z_2, \ldots z_n$ denotes the vertical height on which there is an image feature located above the grid $(x, y)$, and $f\_list(x, y, z)$ is a list pointing to features in the feature bag. It should be denoted this expression is quite different from traditional methods, as there could be multiple features on a single space point $(x, y, z)$. That seems odd, but necessary in practice. When the camera is moving while SLAM, the image feature of one place is actually changing because of the huge change of visual angles and illuminations. In the history of image processing, researchers have been struggling with such changes and developed various features that are "invariant" to scaling, rotation, transformation and illumination, so that they can track the same place of the object and do more other works. In the expression of the proposed 2.5D map, considering the huge change of visual angles and illuminations while SLAM, the feature in one place may change greatly, as a result, we assume one place could have multiple features.

As shown in Figure 5, a 2.5D Map combing with dense obstacle representation on the 2D grid plane and sparse features representation in 3D space above the plane could be obtained, through the combination and optimization with LiDAR and camera sensors. A list of key-frames with poses and visual words extracted while SLAM processing is also attached to the map.

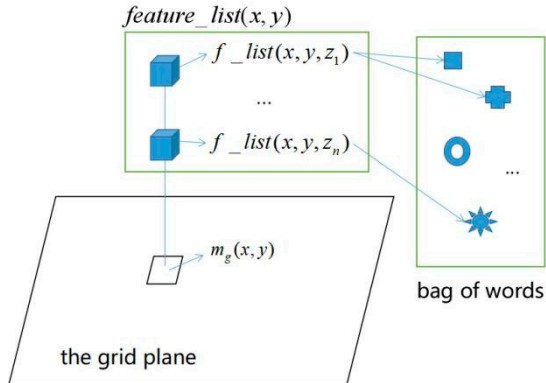

**Figure 5.** The expression of the proposed 2.5D map.

In the map, in order to simplify the calculation, the resolution is uninformed by the grid map, for example, the smallest grid cell could be 5 cm × 5 cm × 5 cm. In that case, there may be multiple feature points with different feature value detected in one cell. That situation is also considered in the map representation.

*4.3. Relocation with Proposed 2.5D Map*

In real applications, although the robot already has the environment map, it may usually lose its location. Such a situation happens when the robot starts on an unknown position, or was kidnapped by a human (blocking the sensors and carry it away) while working. Relocation is required, and its speed and the successful rate have a great influence on the practicability of a mobile robot system.

Currently, for the grid map created by LiDAR-SLAM approaches, Montecalo and particle filter-based method is wildly applied to find the robot pose. However, because the scan data carries too little unique information of the environment, it may take a long time for the robot to find its location. On the other hand, image information is rich enough for fast place finding. In this paper, with the aid of image features and BoW, a fast relocation algorithm with our map is as follows:

Firstly, extract current image features, calculate the bag-of-features (visual words) with BoW; Secondly, List all the previous key-frames with poses in map which shares visual words with current image, and find *n* best match key-frames with ranking score through BoW searching; Thirdly, for *n* best match key-frames, let their poses as initial guess for particles; Finally, apply particle filter based method to find best robot pose, the error function could be formula (6) in Section 3.

## 5. Experiment

The experiment is divided into three parts. In the first part, a comparative experiment of fixed-point positioning accuracy is carried out in a small range of scenes. The traditional laser SLAM based on graph optimization, the visual SLAM based on orb feature point extraction and the laser vision method proposed in this paper is used to collect positioning data. In the second part, a large scene loop experiment is carried out to verify whether the proposed method can effectively solve the problem of map closure in laser SLAM. In the last part, we load the built map for the re-localization experiment.

### 5.1. Experimental Platform and Environment

The experiment was carried out on a robot platform based on Turtlebot 2, equipped with a notebook, a LiDAR and an RGB-D camera. The notebook has Intel Core i5 processor and 8G memory, running on Ubuntu 14.04 + ROS Indigo system. The robot platform is shown in Figure 6:

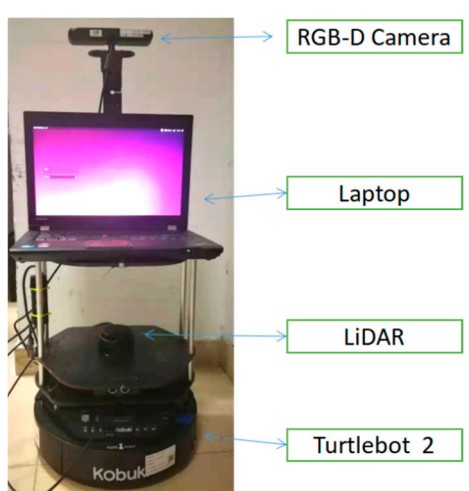

**Figure 6.** The robot platform for the experiment.

The LiDAR is RPLIDAR-A2 produced by SLAMTEC company. It is a low-cost LiDAR for service robotics, which has a 360° coverage field and range of up to 8 m. The key parameters are listed in Table 1. It should be noted that the parameters are taken under the ideal situation, and being a low-cost LiDAR, the data collected from it is usually much poorer than expensive ones in a real scene.

**Table 1.** The key parameters of RPLidar-A1.

| Measuring Distance | Ranging Accuracy | Angle Range | Angle Resolution | Frequency |
|---|---|---|---|---|
| 0.15–12 m | 0.001 m | 0–360° | 0.9° | 10 |

The RGB-D camera is Xtion-pro produced by ASUS company. The effective range of depth measurement is 0.6–8 m, the precision is 3 mm, and the angle of view of the depth camera can reach horizontal 58° and vertical 45.5°.

With the robot platform, we have collected several typical indoor databases in the "rosbag" format, which is easy to read for ROS. For each database, it contains sensor data obtained by the robot while it is running in a real scene, including odom data, laser scan, color image and depth image. Figure 7 shows

one example of the database. Where Figure 7a displays the robot in the environment; Figure 7b shows the scan data obtained in the place; Figure 7c is the RGB image; Figure 7d is the depth image.

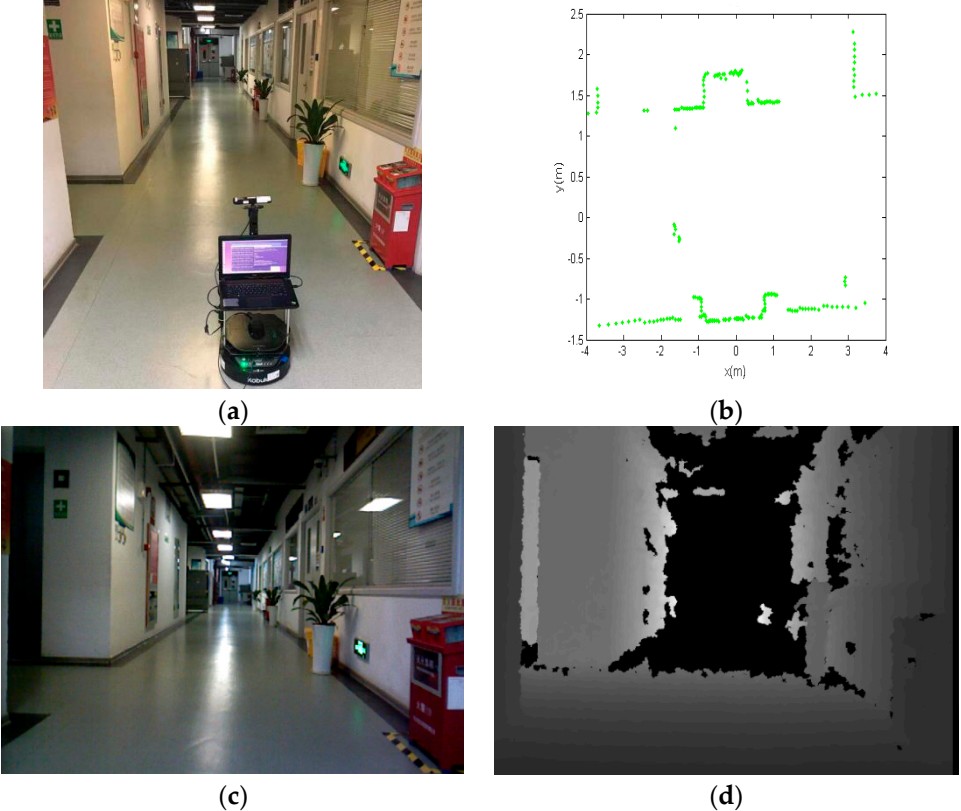

**Figure 7.** One example of the database: (**a**) The robot in the environment; (**b**) scan data obtained in the scene; (**c**) the RGB image; (**d**) the depth image.

Different experiments were taken to evaluate the performance of mapping and relocation. For the mapping, the methods for comparison include Karto-SLAM [3], orb-SLAM [9] and this paper. For the relocation with s map, we compared our method with the Adaptive Monte Carlo Localization (AMCL) [34]. In the experiments, the parameter $\beta$ of formulas (6) and (8) is set to 0.2, by considering the precision and reliability of the two main sensors in the environment.

## 5.2. Experiment of Building the Map

Firstly, in order to evaluate the positioning accuracy while mapping, we manually marked 6 positions in a real scene, as shown in Figure 8. The start point position 0 is specified as the original point (0,0) in the world coordinate system. As shown in Figure 8, the robot started from position 0 and stopped at position 5, and the blue arrow in each picture is the robot's moving direction. Table 2 shows the result of positioning in the SLAM system from position 1 to position 5.

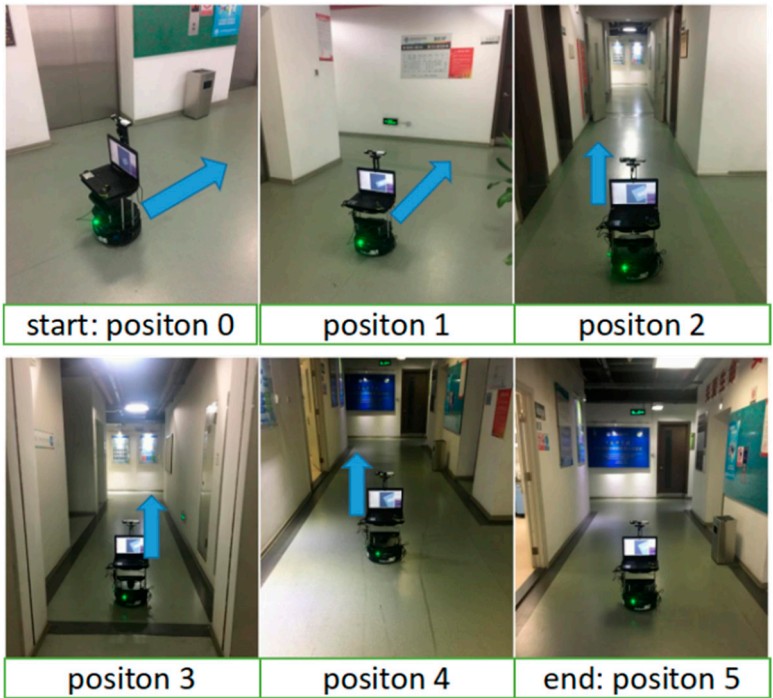

**Figure 8.** The marked positions of the robot.

As can be seen from Table 2, for both methods, the initial error of positioning is small. However, while the robot was running further, the error increases with distance. Sooner or later, such error will be too large to be ignored, and finally cause failure to mapping. Such a situation may become worth for a low-cost LiDAR. That's why we need a loop closing module for thr SLAM system.

**Table 2.** The positioning result while Simultaneous Localization and Mapping (SLAM).

| Position Number | Real Position (m) | Karto-SLAM (m) | Our Method (m) |
| --- | --- | --- | --- |
| 1 | (3,0) | (2.995,−0.002) | (2.994,−0.003) |
| 2 | (6,0) | (5.987,−0.003) | (5.990,−0.005) |
| 3 | (6,−8) | (6.025,−8.032) | (6.019,−8.021) |
| 4 | (6,−16) | (6.038,−15.954) | (6.028,−15.973) |
| 5 | (3,−16) | (2.949,−15.946) | (2.951,−15.965) |

We did further mapping experiment in a real scene compared with Karto-SLAM and Orb-SLAM. The result is shown in Figure 9. Figure 9a is the result of Karto-SLAM, the blue points are the estimated robot position during SLAM. Because of the growing cumulative error, the starting point and the ending point in the red coil cannot be fused together, the mapping failed. Figure 9b is the feature map constructed by orb-SLAM. It should be denoted that orb-SLAM is easy to lose tracking, and the robot should run and turn very slow while building the map. When tracking is lost, it needs a manual operation to turn back to find previous key-frames. As a result, orb-SLAM can't work directly with our recorded rosbag databases. Figure 9c is the grid map part of the proposed method, the green points are the robot positions estimated during SLAM. Because of the effective loop detection and optimization, the map is much better than compared methods. The final 2.5D map showing both grid and feature point is shown in Figure 10.

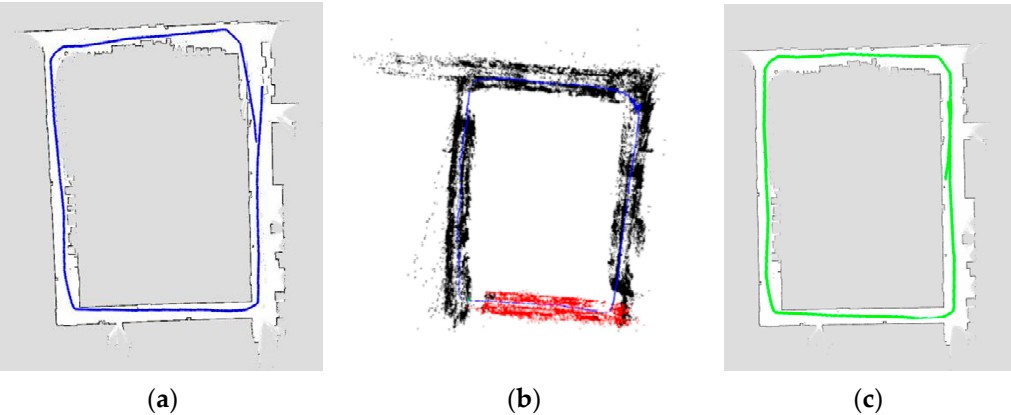

**Figure 9.** Comparison of map constructed by different methods: (**a**) the result of Karto-SLAM; (**b**) the result of orb-SLAM; (**c**) the result of the proposed method (showing only grid map).

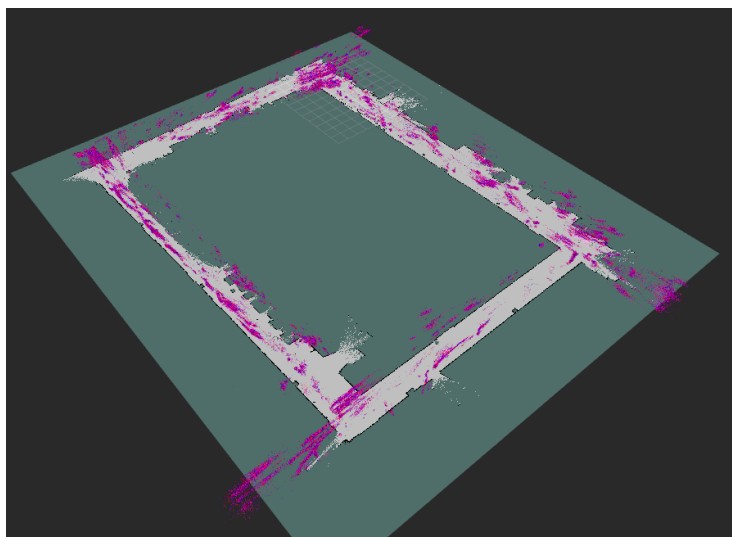

**Figure 10.** The result of the proposed method showing feature position and grid map.

*5.3. Experiment of Relocation*

Finally, with the 2.5D map build by the proposed method, relocation experiment is carried out. The comparing methods are the Adaptive Monte Carlo Localization method (with LiDAR and grid map) and orb-SLAM localization mode (with camera and feature map).

From the practical point of view of the robot, when the position is lost and after the interference is removed, the shorter the relocation time of the robot is, the better. which is also the reason why rapid relocation is needed. In our experiment, considering the practicability and convenience of the robot, we artificially set the relocation time threshold to 30 s.

During the experiment, the robot is put in a real scene in random positions, with or without manually given initial poses. With running different localization algorithms, the robot is driven randomly in the environment until it finds its correct location. Finally, if time exceeds the given threshold (30 s) or converges to a wrong position, we judge it as an unsuccessful relocation try. The successful rate of relocation and average time consumed (if succeed) are shown in Table 3.

**Table 3.** The relocation results of a different method.

| method | Map and Sensor | With Given Initial Pose | | Without Given Initial Pose | |
|---|---|---|---|---|---|
| | | Success Rate | Avg. Time(s) | Success Rate | Avg. Time(s) |
| AMCL [34] | Grid map with LiDAR | 95% | 6.2 | 30% | 20.8 |
| Orb-SLAM Localization [9] | Feature map with camera | - | - | 88% | 5.8 |
| Proposed method | 2.5D map with LiDAR and Camera | 95% | 8.7 | 92% | 6.6 |

As can be seen from Table 3. The AMCL approach needs given initial pose for fast relocation, or it may usually fail in limited time. The Orb-SLAM adopted the bag of words approach, which is capable of fast global relocation, the relocation failure happens in case of a similar scene in vision. Relocation method with our proposed map is faster and higher in successful rate than compared methods, because both laser scan and image are considered.

## 6. Conclusions and Future Work

In this paper, a SLAM framework based on low-cost LiDAR and vision fusion is introduced, as well as a new expression of the 2.5D map and fast relocation method. The results gathered show that with the combination of scan data and image data, the proposed method can improve the mapping and relocation performance compared with traditional methods.

In the future, we will focus on developing and improving Vision-Laser data fusion algorithms to improve the performance of SLAM under more complex and dynamic environments. On the other hand, the proposed 2.5D map could be useful for analyzing the semantics of scene and object. We will also carry out in-depth research in this area.

**Author Contributions:** Conceptualization, S.J., G.J. and L.Y.; methodology, G.J., S.J.; software, G.J., Lei Yin; validation, L.Y., C.T.; formal analysis, S.J.; data acquisition, L.Y.; writing—original draft preparation, G.J.; writing—review and editing, X.M.; supervision, Y.O.; project administration, Y.O.

**Funding:** This work was supported by the National Natural Science Foundation of China (Grants No. U1613210), Guangdong Special Support Program (2017TX04X265), Primary Research & Development Plan of Guangdong Province (2019B090915002), Shenzhen Fundamental Research Program (JCYJ20170413165528221, JCYJ20164428154842603).

**Conflicts of Interest:** The authors declare no conflict of interest.

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
