# Peer review of "A Simultaneous Localization and Mapping (SLAM) Framework for 2.5D Map Building Based on Low-Cost LiDAR and Vision Fusion"

_applsci, doi:10.3390/app9102105_

Round 1
Reviewer 1 Report
The authors describe a method to incorporate LiDAR measurements for SLAM in order to obtain an accurate SLAM system. The authors claim their system is more accurate than LiDAR only or Camera only system.
Reference missing:
"Direct Visual SLAM using Sparse Depth for Camera-LiDAR System" by Young-Sik Shin, Yeong Sang Park and Ayoung Kim
They tackle the same problem of using LiDAR measurements in visual SLAM but no citation has been given. Unless the authors describe significant difference with the mentioned work, it cannot be accepted.
Several aspects of writing need a lot of improvements, there are too many to write here.
Equations have problem: eq. (7), eq. (8) and eq. (9). The hat seems to have been mistaken for a carat sign "^". "K" seems to be have been redefined at some point from its original definition of camera parameters.
Eq. (14)-- (20) are completely untied to the original cost. It does not benefit to put these equations just to add more lines to the text.
Propose instead of "purpose" - abstract and the rest.
"Reference [22]" can be changed to [22].
Author Response
Response to Review-1
Thank you for the time and efforts on our paper and for providing helpful comments and suggestions. We have seriously considered these comments and suggestions, and carefully revised the manuscript accordingly. We marked the revised parts with the "Track Changes" function to make them easy to identify.
Following are the responses to comments:
Comments 1:
”Reference missing:
"Direct Visual SLAM using Sparse Depth for Camera-LiDAR System" by Young-Sik Shin, Yeong Sang Park and Ayoung Kim.
They tackle the same problem of using LiDAR measurements in visual SLAM but no citation has been given. Unless the authors describe significant difference with the mentioned work, it cannot be accepted.“
Response 1:
Thank you very much for reminding us the latest related work of Young-Sik Shin. We have downloaded the paper and studied it seriously.
Generally speaking, their paper tackle the same problem of Camera-LiDAR SLAM and. However, there are several significant differences:
1, Sensor system and applications. Their paper applies mono camera and 3D Lidar, which is mainly used in outdoor robots or unmanned vehicles. While our paper adopts the RGB-D camera and low-cost 2D Lidar, for indoor robot systems.
2, Difference in sensor fusion. Their paper proposes direct visual SLAM only when sparse depth measurements are available, in other words, they track the point in space which have both RGB data from camera and Depth data from Lidar. While in our paper, we do not need the camera and Lidar to observe the same area or object at the same time, an error function is introduced to unite this two measurements.
3, Different map. Their paper constructs 3D point cloud map; while our paper constructs 2.5D map.
We have added their paper in our introduction part.
We also did major revision to the section ‘1.3. Multi-Sensor Fusion’. Following are the added reference papers.
1. López, E.; García, S.; Barea, R.; Bergasa, L.M.; Molinos, E.J.; Arroyo, R.; Romera, E.; Pardo, S. A Multi-Sensorial Simultaneous Localization and Mapping (SLAM) System for Low-Cost Micro Aerial Vehicles in GPS-Denied Environments. Sensors 2017, 17, 802. doi: 10.3390/s17040802
2. Nam, T.H.; Shim, J.H.; Cho, Y.I. A 2.5D Map-Based Mobile Robot Localization via Cooperation of Aerial and Ground Robots. Sensors 2017, 17, 2730. doi: 10.3390/s17122730
3. Zhang, Z.; Zhao, R.; Liu, E.; Yan, K.; Ma, Y. Scale Estimation and Correction of the Monocular Simultaneous Localization and Mapping (SLAM) Based on Fusion of 1D Laser Range Finder and Vision Data. Sensors 2018, 18, 1948. doi: 10.3390/s18061948
4. Shin Y.; Park Y.; Kim A. "Direct Visual SLAM Using Sparse Depth for Camera-LiDAR System," 2018 IEEE International Conference on Robotics and Automation (ICRA), Brisbane, QLD, 2018, pp. 1-8.
5. Xu Y.; Ou Y.; Xu T. "SLAM of Robot based on the Fusion of Vision and LIDAR," 2018 IEEE International Conference on Cyborg and Bionic Systems (CBS), Shenzhen, 2018, pp. 121-126.
Comments 2:
“Several aspects of writing need a lot of improvements, there are too many to write here.”
“Equations have problem: eq. (7), eq. (8) and eq. (9). The hat seems to have been mistaken for a carat sign "^". "K" seems to be have been redefined at some point from its original definition of camera parameters.”
Response 3:
The equations have been corrected.
Camera parameter ‘K’ is redefined to ‘C’.
Comments 4:
“Eq. (14)-- (20) are completely untied to the original cost. It does not benefit to put these equations just to add more lines to the text.”
Response 4:
We did major revision to the section ‘3.2. Pose graph optimization’, from line 285 to line 304. To make a more clear expression.
Comments 5:
“Propose instead of "purpose" - abstract and the rest.”
Response 5:
All 23 such misspellings have been corrected.
Comments 6:
"Reference [22]" can be changed to [22].
Response 6:
We did major revision to the section ‘1.3. Multi-Sensor Fusion’, and added 5 more related works.

Reviewer 2 Report
Summary
This present manuscript purposes a robust low-cost Simultaneous Localization and Mapping frame work, with the combination of low-cost Light Detection and Ranging, and vision. By combining the laser points cloud data and image feature points data as constraints, an graph optimization method is used to optimize the pose of the robot. At the same time, the Bag of Words based on visual features is used for loop closure detection, and then the grid map built by laser is further optimized. Finally, a 2.5D map presenting both obstacles occupy and vision feature is built
Broad Comment
The present manuscript presents an interesting approach, based on the fusion of two well-known technologies (SLAM and LiDAR). The facts that low cost LIDAR radar is used possess a major challenge that the authors manage obtaining interesting results.
I would suggest the authors to include more recent work done with the same approach, leaving an example:
Y. Xu, Y. Ou and T. Xu, "SLAM of Robot based on the Fusion of Vision and LIDAR," 2018 IEEE International Conference on Cyborg and Bionic Systems (CBS), Shenzhen, 2018, pp. 121-126. DOI: 10.1109/CBS.2018.8612212
Following are some pertinent comments on the answer, not calling into question the work done. Authors should also check for some typos.
Specific Comments
1. Remove lines 14 and 15;
2. Define LIDAR for the first time used (line 16);
3. Check line 25, 53, 108, 188, 197, 305, 403;
4. Define UAV for the first time used (line 88);
5. Define IMU for the first time used (line 102);
6. Define ICP for the first time used (line 107);
7. Gradient methods are used (line 163) but no clear observations are made about the initializations of the methods. This is an important improvement that should be made;
8. Use bold for vector and describe their indices (lines 169, 171, 186, 205, 229, 256, 258);
9. Parameter β, (equation 6 and lines 236, 238 and 239) is presented, a methodology for tuning is described but I don’t think that there is enough information to reproduce the results, since the parameter is never referred again;
10. Add some references for the methods presented on paragraph starting line 275;
11. Considering the mathematical formulation, what would happened if matrix H is not invertible?
12. Define ORB for the first time used (line 325);
13. Why the text in bold (line 371)?
14. Section 5 don’t describe completely all the parameters used, namely β. Please add some information.
15. Define ROS for the first time used (line 431);
16. Table 2 has formatting problems;
17. Lines 492-494, why 30 seconds. How much time would it be needed?

Author Response
Response to Review-2
Thank you very much for your recognition of our work. And the careful work and valuable suggestions while reviewing this paper. We have seriously considered these comments and suggestions, and carefully revised the manuscript accordingly. We marked the revised parts with the "Track Changes" function to make them easy to identify.
Following are the responses to comments:
Comments 1:
“The present manuscript presents an interesting approach, based on the fusion of two well-known technologies (SLAM and LiDAR). The facts that low cost LIDAR radar is used possess a major challenge that the authors manage obtaining interesting results.
I would suggest the authors to include more recent work done with the same approach, leaving an example:
Y. Xu, Y. Ou and T. Xu, "SLAM of Robot based on the Fusion of Vision and LIDAR," 2018 IEEE International Conference on Cyborg and Bionic Systems (CBS), Shenzhen, 2018, pp. 121-126. DOI: 10.1109/CBS.2018.8612212”
Response 1:
We have added 5 more related works (listed below) to the introduction part, and did major revision to the section ‘1.3. Multi-Sensor Fusion’.
1. López, E.; García, S.; Barea, R.; Bergasa, L.M.; Molinos, E.J.; Arroyo, R.; Romera, E.; Pardo, S. A Multi-Sensorial Simultaneous Localization and Mapping (SLAM) System for Low-Cost Micro Aerial Vehicles in GPS-Denied Environments. Sensors 2017, 17, 802. doi: 10.3390/s17040802
2. Nam, T.H.; Shim, J.H.; Cho, Y.I. A 2.5D Map-Based Mobile Robot Localization via Cooperation of Aerial and Ground Robots. Sensors 2017, 17, 2730. doi: 10.3390/s17122730
3. Zhang, Z.; Zhao, R.; Liu, E.; Yan, K.; Ma, Y. Scale Estimation and Correction of the Monocular Simultaneous Localization and Mapping (SLAM) Based on Fusion of 1D Laser Range Finder and Vision Data. Sensors 2018, 18, 1948. doi: 10.3390/s18061948
4. Shin Y.; Park Y.; Kim A. "Direct Visual SLAM Using Sparse Depth for Camera-LiDAR System," 2018 IEEE International Conference on Robotics and Automation (ICRA), Brisbane, QLD, 2018, pp. 1-8.
5. Xu Y.; Ou Y.; Xu T. "SLAM of Robot based on the Fusion of Vision and LIDAR," 2018 IEEE International Conference on Cyborg and Bionic Systems (CBS), Shenzhen, 2018, pp. 121-126.
Comments 2:
“1. Remove lines 14 and 15;
2. Define LIDAR for the first time used (line 16);
3. Check line 25, 53, 108, 188, 197, 305, 403;
4. Define UAV for the first time used (line 88);
5. Define IMU for the first time used (line 102);
6. Define ICP for the first time used (line 107);
12. Define ORB for the first time used (line 325);
15. Define ROS for the first time used (line 431);”
Response 2:
The errors have been corrected.
Comments 3:
“7. Gradient methods are used (line 163) but no clear observations are made about the initializations of the methods. This is an important improvement that should be made;.”
Response 3:
We have revised the part in section ‘3.2. Pose graph optimization’, and discussed about the initialization :
“In our approach, the robot pose set is regarded as the variable to be optimized”
“the initial value of x is obtained mainly through scan matching”
Comments 4:
“8. Use bold for vector and describe their indices (lines 169, 171, 186, 205, 229, 256, 258)”
Response 4:
The errors have been corrected.
Comments 5:
“9. Parameter β, (equation 6 and lines 236, 238 and 239) is presented, a methodology for tuning is described but I don’t think that there is enough information to reproduce the results, since the parameter is never referred again”
Response 5:
In our current work, the is given manually. In line 441, we introduce the in our work:
“In the experiments , the parameter of formula (6) and (8) is set to 0.2, by considering the precision and reliability of the two main sensors in the environment.”
We will do follow-up work in the future to the auto adjustment of .
Comments 6:
"10. Add some references for the methods presented on paragraph starting line 275;.
Response 6:
In line 282, the following references are added.
1. Bjorck, A. (1996) Numerical Methods for Least Squares Problems. SIAM, Philadelphia.
2. Morrison, David D. "Methods for nonlinear least squares problems and convergence proofs". Proceedings of the Jet Propulsion Laboratory Seminar on Tracking Programs and Orbit Determination, 1960: 1–9.
Comments 7:
"11. Considering the mathematical formulation, what would happened if matrix H is not invertible?”
Response 7:
Because H is calculated by JTJ here, it contains the constraints between poses, and only the adjacent posed have direct constraints, most of the elements are 0. H is sparse and shaped like an arrow. Most of the time, H is invertible. If it is not invertible, in formula (17) will make the left part invertible, and that’s why L-M method is more stable.
We did major revision from line 285 to line 304, to to make the expressions more clearly. In line 293, we made a discussion about this:
“As L-M method could be regarded as a combination of Gadient Descent method and Gauss Newton method in the formula (17), it could be more robust to get a reliable even if is not invertible. “
Comments 8:
"13. Why the text in bold (line 371)?”
Response 8:
It was meant to emphasize this fact. Bold format is removed now.
Comments 9:
“14. Section 5 don’t describe completely all the parameters used, namely β. Please add some information.”
Response 9:
In line 441, we introduce the in our work:
“In the experiments , the parameter of formula (6) and (8) is set to 0.2, by considering the precision and reliability of the two main sensors in the environment.”
Comments 10:
“Table 2 has formatting problems;”
Response 10:
The errors have been corrected.
Comments 11:
“Lines 492-494, why 30 seconds. How much time would it be needed?”
Response 11:
We added the discussion about this in line 482:
“From the practical point of view of the robot, when the position is lost and after the interference is removed, the shorter the relocation time of the robot is, the better. which is also the reason why the rapid relocation is needed. In our experiment, considering the practicability and convenience of the robot, we artificially set the relocation time threshold to 30 seconds.

Round 2
Reviewer 1 Report
I find the new version sufficiently improved and many of the reviewers’ concerns have been addressed. I advise one more round of corrections in overall writing and grammar before going for the publication.